# Did a bot eat your homework? An assessment of the potential impact of bad actors in online administration of preference surveys

Juan Marcos Gonzalez[1]*, Kiran Grover[2], Thomas W. Leblanc[1,3], Bryce B. Reeve[1]

1 Department of Population Health Sciences, Duke University School of Medicine, Durham, North Carolina, United States of America, 2 Icahn School of Medicine, Mount Sinai, New York, New York, United States of America, 3 Department of Medicine, Division of Hematologic Malignancies and Cellular Therapy, Duke University School of Medicine, Durham, North Carolina, United States of America

* jm.gonzalez@duke.edu

## Abstract

**Data Availability Statement:** All relevant data are within the manuscript and its Supporting information files.

### Background

Online administration of surveys has a number of advantages but can also lead to increased exposure to bad actors (human and non-human bots) who can try to influence the study results or to benefit financially from the survey. We analyze data collected through an online discrete-choice experiment (DCE) survey to evaluate the likelihood that bad actors can affect the quality of the data collected.

### Methods

We developed and fielded a survey instrument that included two sets of DCE questions asking respondents to select their preferred treatments for multiple myeloma therapies. The survey also included questions to assess respondents' attention while completing the survey and their understanding of the DCE questions. We used a latent-class model to identify a class associated with perverse preferences or high model variance, and the degree to which the quality checks included in the survey were correlated with class membership. Class-membership probabilities for the problematic class were used as weights in a random-parameters logit to recover population-level estimates that minimizes exposure to potential bad actors.

### Results

Results show a significant proportion of respondents provided answers with a high degree of variability consistent with responses from bad actors. We also found that a wide-ranging selection of conditions in the survey screener is more consistent with choice patterns expected from bad actors looking to qualify for the study. The relationship between the number of incorrect answers to comprehension questions and problematic choice patterns peaked around 5 out of 10 questions.

**Funding:** Financial support for this study was provided in part by Amgen, Inc. The funding agreement ensured the authors' independence in designing the study, interpreting the data, writing, and publishing the report.

**Competing interests:** The authors have declared that no competing interests exist.

## Conclusions

Our results highlight the need for a robust discussion around the appropriate way to handle bad actors in online preference surveys. While exclusion of survey respondents must be avoided under most circumstances, the impact of "bots" on preference estimates can be significant.

## 1. Introduction

Online administration of surveys is a common way to collect preference data for a variety of reasons. These include: 1) the online platforms' support for complex design features and adaptive presentation of the survey content, 2) scalability of data collection, 3) relatively broad access of participants to the internet either via computer or (increasingly) cell phones, 4) the anonymity enjoyed by respondents can reduce social desirability bias in their responses, 5) convenience for respondents, 6) elimination of manual data entry, and 7) the overall speed of data collection [1].

Unfortunately, this type of administration can also have significant potential drawbacks. Some well-known issues with online administration include: 1) potentially skewed distribution of respondents, as online respondents tend to be younger, with lower income levels, less racially/ethnically diverse and more highly educated than the average population [2], 2) higher attrition rates, 3) increased chance of multiple responses from single individuals, and 4) lack of control over environmental stimuli or distractions [3].

Another potential issue with online administration is the increased exposure to bad actors who can try to influence the study results or to benefit financially from the survey. These actors could use computer programs (bots) to quickly complete many surveys or repeatedly complete the surveys themselves. The latter have been termed "human bots" and differ from inattentive respondents in that the former participants misrepresent their eligibility and do not aim to provide any meaningful information [4]. These drawbacks are not new and are not strictly synonymous with online administration. Online panels can and have developed ways to actively manage membership and offline administration of surveys can still be subject to some of the issues outlined here. Nevertheless, the evolution of new and more sophisticated artificial intelligence tools that can be leveraged by bad actors make these issues potentially more problematic with online administration.

While evidence suggests that members of online panels and crowdworkers (e.g., MTurkers) can provide reasonable evidence on patient preferences [5, 6], bots represent a significant threat to scientific research as bot-generated data are not valid [7]. To date, there is no clear sense of the prevalence of bot data in academic research, but experts agree that their presence is probably on the rise [8]. Given that about 40% of all internet use is attributed to bots, concerns about the use of bots to complete preference surveys seems warranted.

The limited evaluation of the impact of online administration on health-preference research has largely focused on process factors like the device used to access the survey [9, 10] and the specific capabilities of the medium, such as the ability to use videos to present the information stimuli [11]. Some preference research, particularly outside of health, has done more formal evaluations on sampling issues associated with online administration, including online panels [12–15]. These include evaluations of the representativeness of the sampling frame, representativeness of the sample, and the representativeness of survey completion.

Our work analyzes data collected through an online discrete-choice experiment (DCE) survey to evaluate the likelihood that bad actors, human or non-human bots, can affect the quality

of the data collected. We used information collected outside of the DCE questions to inform the likelihood that respondents provided bad preference information. The preference data were collected to document preferences from patients with multiple myeloma (MM) in the United States. Specifically, to understand patients benefit-risk preferences associated with treatment choices among patients who are currently taking second-line therapy (and beyond) for MM.

Multiple myeloma is a rare disease with an estimated world-wide 5-year prevalence of 230,000 patients [16]. Treatment options for multiple myeloma patients have been increasing steadily in recent years, offering patients different levels of efficacy and adverse events. Matching these patients with treatments that fit their preferences can help minimize the burden of these treatments and their disease. Conducting research that can help inform these decisions requires reaching an adequate number of patients to collect their perspectives on the matter, but this entails tradeoffs between ease of access to available patients and potential exposure to bad actors. Such tradeoffs make it critical to have ways to ascertain the degree to which we can be confident that the information collected is indeed from patients. Our work sought to evaluate the strategies implemented by the study team and to quantify their association with the elicitation of meaningful preference information.

## 2. Methods

We developed and fielded a survey instrument that included two DCE modules following good practices for health preference research [17, 18]. In each DCE module, respondents were asked to select their preferred treatment alternatives in a series of experimentally-controlled pairs of MM therapies. Respondents had to be residents of the United States (US) with self-reported physician diagnosis of MM and with previous history of treatment discontinuation. They also had to be able to understand English and provide their consent to participate in the study. The study was reviewed and approved by the Institutional Review Board at Duke University. Verbal informed consent was collected from respondents who participated in interviews to test the survey instrument. Online participants provided electronic consent (e-consent) before accessing the survey.

The alternatives evaluated in the DCE modules were defined based on common features (attributes) and the performance of each alternative under each attribute (attribute levels) and were defined after a series of concept elicitation interviews with 21 patients with MM. Interviewed patients were recruited through a large medical center in the United States and the MM registry of the Cancer Support Community. The interviews focused on patients' experiences with MM and its treatments, and looked to collect evidence on the most salient aspects of treatments from patients.

The final attributes in the first DCE module (Table 1) considered efficacy (chance of complete response and progression-free survival), treatment-related toxicities (gastrointestinal problems, neurotoxicity, and insomnia), and treatment-related risks of adverse events (cytokine release syndrome [CRS]). An example choice question is presented in Fig 1.

The second module was used to ascertain patient preferences for novel MM therapies. Namely, Chimeric antigen receptor (CAR-T) therapy, bi-specific antibody (presented as BiTE® which is a proprietary version of the technology in this class and stands for bi-specific T-cell engager) and Anti-drug conjugates (ADC). We prepared videos that explained the three novel MM therapies included in module 2. These videos lasted 7 minutes in total and respondents were not allowed to skip them. Table 2 summarizes the attributes and attribute levels included in the survey instrument and Fig 2 shows an example question from the second module. It is important to note that how the medicine was taken and the type of side effects that

**Table 1. Attributes and levels in module 1.**

| Effect category | Attribute | Levels |
|---|---|---|
| Treatment type | Route of administration | • Oral<br>• Injection<br>• Intravenous |
| Benefits | Chance of complete response | • 50 out of 100 (50%)<br>• 20 out of 100 (20%)<br>• 0 out of 100 (0%) |
| | Progression-free survival (time until cancer gets worse) | • 30 months<br>• 18 months<br>• 12 months |
| Toxicities | Gastrointestinal problems | • No stomach problems<br>• Severe diarrhea<br>• Severe nausea |
| | Neurotoxicity | No problems with the nervous system<br>• Peripheral neuropathy<br>• CNS neurotoxicity |
| | Insomnia | • No insomnia<br>• Mild insomnia<br>• Severe insomnia |
| Adverse-event risks | Chance of cytokine release syndrome (CRS) | • 0 out of 100 (0%)<br>• 3 out of 100 (3%)<br>• 5 out of 100 (5%) **or** 10 out of 100 (10%) |

CNS = Central nervous system.

could be experienced in the second module were directly associated with one type of treatment and were not independently designed in the experiment. Thus, these attributes are not included in Table 2 despite being shown in Fig 2.

Respondents were randomized to consider choice questions in which the highest levels of a specific toxicity (CRS) were 5% or 10% to test for potential re-coding of attribute levels to an ordinal scale [19]. Both modules presented the same highest level of CRS to respondents once assigned.

To prepare respondents for the choice questions, the survey instrument included carefully worded descriptions of all attributes, comprehension questions, a tutorial for probabilistic attributes, and simplified practice choice questions. One-on-one pretest interviews with 10 MM patients provided important insights regarding respondents' reactions to the survey instrument. Interviewers followed a think-aloud protocol in which respondents were asked to read the survey instrument out loud and encouraged to articulate their thoughts related to survey information materials and questions. During the interviews, respondents completed an online version of the survey using their personal computers at home. They answered probe questions from interviewers to assess their use of survey information and graphics. Bidding games were used to ascertain respondents' willingness to accept tradeoffs among survey attributes and to check for internal consistency of responses. At the end of the pretest interviews, we found that respondents were able to complete the survey without any issues and to provide valid answers to all questions.

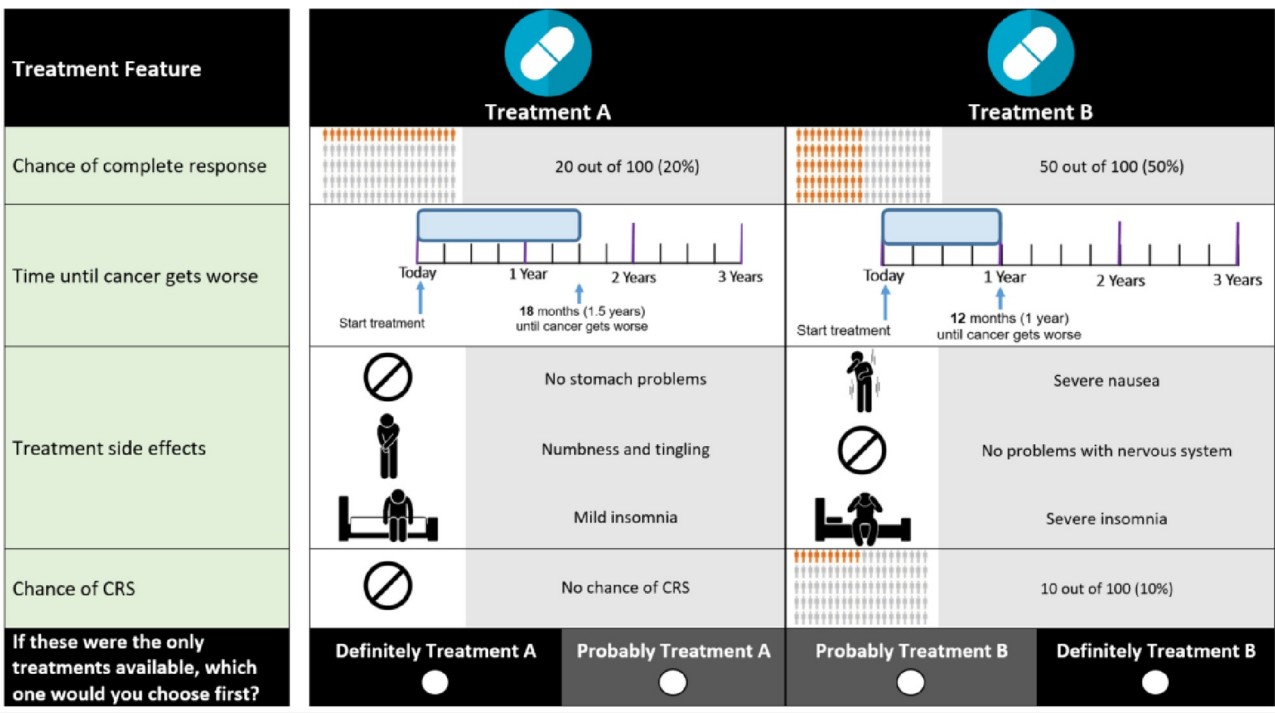

**Fig 1. Example choice question in module 1.**

Upon finalization of the pretest interviews, a final version of the survey was programmed and an experimental design of the choice questions was prepared. The experimental design determines the combinations of attribute levels that define each hypothetical treatment profile and to pair of hypothetical therapies that will populate the DCE choice questions.

The experimental designs for the two DCE modules were prepared to maximize the statistical efficiency (D-optimality) of the choice questions without priors or constraints. SAS 9.4 (Cary, NC) [20] was used to generate several design options with very similar statistical efficiency. Each of these designs was evaluated to assess level balance [21]. Two experimental designs were

**Table 2. Attributes and levels in module 2.**

| Effect category | Attribute | Levels |
|---|---|---|
| Treatment type | | • CART-T therapy<br>• BiTE® therapy<br>• ADC therapy |
| Benefits | Chance of complete response | • 50 out of 100 (50%)<br>• 20 out of 100 (20%)<br>• 0 out of 100 (0%) |
| Adverse-event risks | Chance of cytokine release syndrome | • 0 out of 100 (0%)<br>• 3 out of 100 (3%)<br>• 5 out of 100 (5%) **or** 10 out of 100 (10%) |

CAR-T = Chimeric Antigen Receptor; BiTE® = Bispecific T-Cell Engager; ADC = Anti-Drug Conjugate.

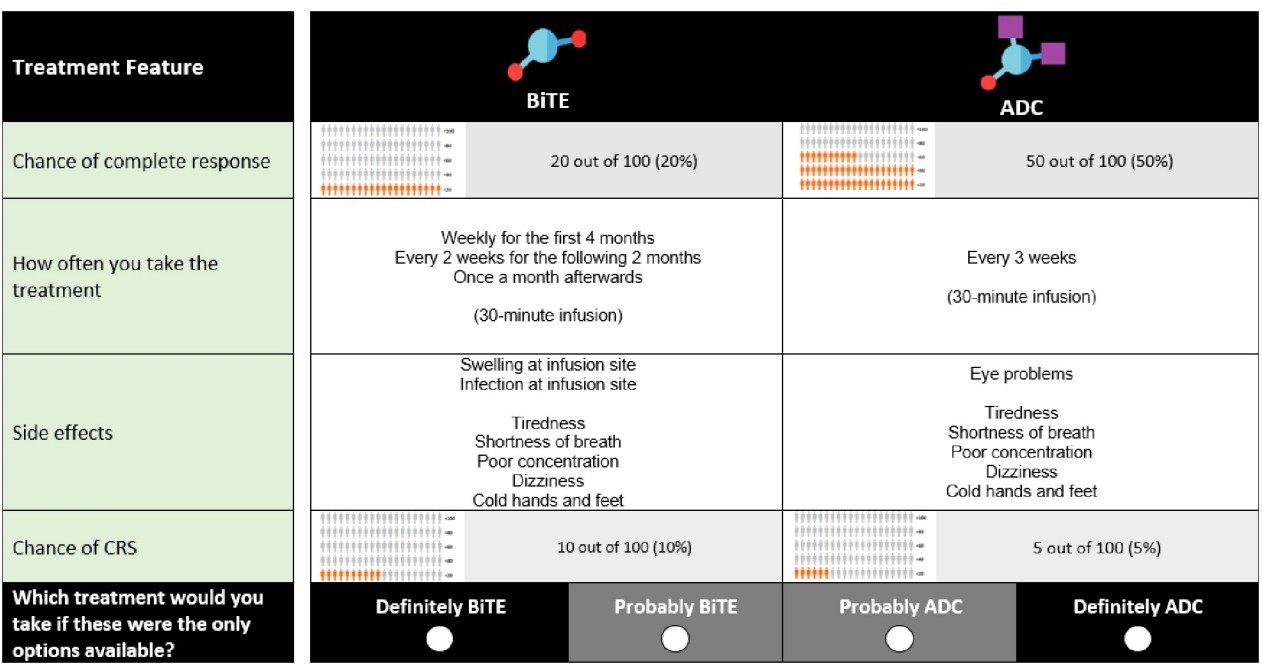

**Fig 2. Example choice question in module 2.**

selected, one with 36 questions for module 1 and another design with 18 questions for module 2. To reduce the overall survey burden, we divided the designs into blocks of 6 and 3 questions, respectively and only one block from each design was randomly assigned to each respondent. Thus, respondents were only asked to answer a total of 9 choice questions. Based on patient feedback during the pretest interviews, respondents were expected to be able to answer these many choice questions before becoming overly fatigued. The study team used response simulations to evaluate the statistical properties of the random assignment of the two designs.

## 2.1 Identifying problematic responses

In addition to the DCE questions, the survey included 10 survey comprehension questions to test respondents' understanding of specific elements of information that was critical to complete the DCE in a reliable manner. Respondents also were asked to report specific aspects of their treatment history in different ways and at different points in time during the survey. Two questions were asked in the survey screener and repeated with a different emphasis at the end of the survey.

We also tried to avoid signaling respondents the answers they had to provide to be eligible to participate in the study. It is not uncommon to embed qualifying responses among a long list of other nonqualifying response options in survey screeners. This reduces the probability that respondents who do not meet the inclusion criteria are able to provide the qualifying response by chance. Respondents were asked to select among 10 conditions for which they have received a physician diagnosis. The list included the following conditions with a wide range of prevalence rates (in parentheses); anemia (5.6% [22]), arrhythmias (between 1.5% and 5% [23]), arthritis (22.7% [24]), chronic obstructive pulmonary disease (4.5% [25]), diabetes (13.0% [26]), hypertension (29.0% [27]), kidney disease (14.2% [28]), migraines (11.7%

among adults [29], and obesity (36.5% among adults [30]). We allowed respondents to select all the conditions that applied to them.

## 3. Analysis

Before analyzing the choice data from the DCE modules, we evaluated responses to the 10 comprehension questions in the survey. We also evaluated time to completion, response non-variance, consistency of respondents' reports, and the likelihood of false reports in the screening questions. We assumed that reports were more improbable the longer the list of conditions selected in the screening question. Finally, we evaluated the proximity of the sample age and race to the known characteristics of the MM population as a way to identify potential issues with the representativeness of the sample.

DCEs generate complex cross-section/time-series choice data for each respondent. These data include a dichotomous dependent variable and are analyzed using advanced statistical methods. The basis for the analysis was the model specification assumed when generating the experimental design prior to survey implementation. That specification considered a categorical main-effects model for all study attributes. The statistical analysis of choices provided a measure of the impact of changes in the attribute levels on the likelihood that treatments are selected by respondents. We call these measures attribute-level preference weights [31]. We formally tested for differences in preferences across modules by estimating interaction terms indicating changes in preferences for chance of complete response and CRS across modules—the only attributes repeated across modules.

For our initial analyses of choice data, we used a latent-class (LC) logit model with effect-coded variables for all levels in each attribute. The LC logit model used information from repeated choices for each respondent to classify them probabilistically into one of several classes of patients. The model uses a data-driven process to group respondents based on the similarity of their choices, so within-group variation is smaller than across-group variation. For each class, we estimated a set of preference weights indicating the relative importance of changes in the attribute levels [32]. In addition, the LC model was used to estimate the impact of respondents' characteristics on the likelihood of class membership. Furthermore, we evaluated whether one of the classes was associated with perverse preferences or high model variance, and the degree to which the quality checks included in the survey were correlated with membership to these classes.

Upon identifying a class that was associated with problematic responses, we estimated a random-parameters logit (RPL) weighted by the probability that respondents were not in the problematic respondent class. The RPL provided a set of overall population preference estimates after limiting the influence of problematic responses to the DCE questions. The RPL assumed that preference estimates were normally distributed across respondents and included a scale control variable to account for differences in the absolute values of the preference estimates in the two survey modules [33].

## 4. Results

A total of 350 respondents were recruited by an international online consumer panel to complete the survey online and completed the survey. Respondents were invited to complete the survey based on prior information that suggested they would be eligible to participate. These respondents were offered an incentive equivalent to approximately $8.

Table 3 presents a summary of the general characteristics of the respondents in the sample.

Since the treatment alternatives in the choice questions are randomly assigned to first and second positions, there is no systematic relationship between the alternative placement and the

**Table 3. Summary of general characteristics in the sample.**

| | Statistic or Category | All Respondents (N = 350) |
|---|---|---|
| **All respondents** | | |
| Age (in years) | Mean (SD) | 42.71 (10.27) |
| Gender | Female | 168 (48%) |
| | Male | 182 (52%) |
| | Other or prefer not to answer | 0 (0%) |
| Marital Status | Single/never married | 25 (7.14%) |
| | Married/living as married | 317 (90.17%) |
| | Divorced or separated | 6 (1.71%) |
| | Widowed/surviving partner | 1 (0.29%) |
| | Other | 1 (0.29%) |
| Ethnicity | Hispanic, Latino or Spanish Origin | 57 (16.29%) |
| | Not Hispanic, Latino or Spanish Origin | 293 (83.71%) |
| Racial groups (Select all that apply) | American Indian or Alaskan Native | 5 (1.43%) |
| | Asian | 5 (1.43%) |
| | African American | 22 (6.29%) |
| | Native Hawaiian or Other Pacific Islander | 1 (0.29%) |
| | White | 331 (94.57%) |
| | Other | 3 (0.86%) |
| Education | Some high school | 3 (0.86%) |
| | High school or equivalent | 13 (3.71%) |
| | Some college but no degree | 22 (6.29%) |
| | Technical school | 4 (1.14%) |
| | Associate's degree or 2-year college degree | 30 (8.57%) |
| | 4-year college degree | 116 (33.14%) |
| | Some graduate school but no degree | 7 (2.00%) |
| | Graduate or professional degree | 155 (44.29%) |
| Employment (Select all that apply) | Employed with hourly pay full time | 185 (52.86%) |
| | Employed with salary full time | 107 (30.57%) |
| | Employed with hourly pay part time | 6 (1.71%) |
| | Employed with salary part time | 4 (1.14%) |
| | Self-employed | 14 (4.00%) |
| | Homemaker | 3 (0.86%) |
| | Student | 2 (0.57%) |
| | Retired | 15 (4.29%) |
| | Volunteer work | 0 (0%) |
| | Other | 0 (0%) |
| | Not working but looking for a job | 3 (0.86%) |
| | Not working and NOT looking for a job | 8 (2.29%) |
| | Unable to work or on disability | 7 (2%) |

Note 1: Percentages do not include missing responses in the denominator

Note 2: Percentages may not add to 100% when participants were allowed to select more than one answer.

attribute levels shown in each question. The probability that the preferred treatment alternative would appear in the same position for all 9 questions is less than 0.01%. Therefore, we inferred that respondents who always selected alternatives in the same position were not attentive to the content of the choice questions. Among respondents in the survey, we identified 27 who

always selected the same alternative with the same level of confidence. These respondents were eliminated from the study sample for all modeling of DCE data.

Among the remaining 323 respondents the average patient was 42.84 years of age (SD 10.35). Also, there were 305 White Americans and 22 African American respondents. These results are dramatically different to the known epidemiology of the disease (less than 1% of the cases are diagnosed among people younger than 35, and most people are diagnosed after the age of 65). Furthermore, multiple myeloma is more than twice as common in African Americans than in White Americans [34].

Based on the pretest interviews, the DCE survey was expected to take 20–30 minutes to complete given the several pages of reading materials, videos explaining the three novel MM therapies, and background questions included in the survey. We evaluated the distribution of survey completion times, which ranged from 7.76 minutes to 130.38 minutes, with a median time of 18.98 minutes.

Table 4 presents the distribution of responses for each of the ten comprehension questions in the survey. The percentage of correct answers to the comprehension questions ranged from 19.8% to 90.4%.

Table 5 presents the frequency of respondents who selected a specific number of options to answer the screening question on the patient's health conditions. About 14% of all respondents selected all conditions in the response list and about 29% selected at least 7 of the 10 options included in the screener. About 40% of the sample selected only one condition (i.e., multiple myeloma).

Table 6 summarizes the responses to the pairs of questions repeated in the survey. The table also shows the percentage of respondents who answered them in a particular way. Between 18.9% and 26.1% of respondents changed their answers when they were asked the questions a second time.

The LC results yielded two classes based on model fit and parsimony. These results represent log-odds preference weights [31] and indicate the relative preference for treatments with specific attribute levels, all else equal. Higher preference weights indicate greater chance of choosing a treatment with specific characteristics, lower preference weights indicate lower chance of choosing a treatment with specific characteristics. Figs 3 and 4 present the preference weights for Class 1 and Class 2, respectively. The average class-membership probabilities were 72.1% for Class 1 and 28.9% for Class 2. Estimates from this model are included in S1 Appendix.

The preference results from the two classes suggested that one group of respondents had patterns of choices that were not easily explainable by our model specification (Class 2). The large confidence intervals around estimates for Class 2 indicates a much larger model variance for respondents in this class. Lack of significance in the estimates and perverse mean values for some of these attribute levels (e.g., increased preference for neurotoxicity or lower chance of response) suggest these respondents were not providing data of enough quality to discern preferences on the outcomes presented. Responses among members of Class 1 are generally ordered—better clinical outcomes are associated with higher preference weights—and had tighter confidence intervals, implying a smaller model error.

To estimate the impact of respondents' characteristics on the likelihood of class membership, four continuous variables, and two dichotomous variables were included to explain class membership based on the following factors:

1. Respondent's age (*Age*)

2. Number of diseases selected in the survey screener (*Scrcount*)

**Table 4. Responses to survey-comprehension questions.**

| | Statistic or Category | All Respondents (N = 323) |
|---|---|---|
| How many people achieved complete response after taking Treatment A? | | |
| | 5 out of 100 (5%) | 6 (1.86%) |
| | 10 out of 100 (10%) [CORRECT ANSWER] | 199 (61.6%) |
| | 20 out of 100 (20%) | 86 (26.6%) |
| | 80 out of 100 (80%) | 30 (9.3%) |
| | Don't know / not sure | 2 (0.6%) |
| What kind of treatment is Treatment B? | | |
| | Oral | 91 (28.2%) |
| | Injection | 73 (22.6%) |
| | Intravenous [CORRECT ANSWER] | 152 (47.1%) |
| | Don't know / not sure | 7 (2.2%) |
| How long is Treatment B expected to keep the cancer from getting worse? | | |
| | 12 months | 44 (13.6%) |
| | 18 months | 68 (21.1%) |
| | 24 months [CORRECT ANSWER] | 205 (63.5%) |
| | 48 months | 5 (1.6%) |
| | Don't know / not sure | 1 (0.3%) |
| Which treatment is taken as an injection? | | |
| | Treatment A [CORRECT ANSWER] | 233 (72.1%) |
| | Treatment B | 87 (26.9%) |
| | Don't know / not sure | 3 (0.9%) |
| **Based on what you saw in the video, select True, False or Don't Know/Not sure for each statement:** | | |
| CAR-T therapy requires taking the patient's own T cells and changing them to start attacking cancer cells in the body | True [CORRECT ANSWER] | 292 (90.4%) |
| | False | 27 (8.4%) |
| | Don't know | 4 (1.2%) |
| CAR-T therapy requires using chemotherapy to make space for new T cells | True [CORRECT ANSWER] | 244 (75.5%) |
| | False | 65 (20.1%) |
| | Don't know | 14 (4.3%) |
| **Based on what you saw in the video, select True, False or Don't know/Not sure for each statement:** | | |
| BiTE® therapy modifies the patient's T cell to attack cancer cells in the body | True | 252 (78.0%) |
| | False [CORRECT ANSWER] | 64 (19.8%) |
| | Don't know | 7 (2.2%) |
| BiTE® therapy is taken only once | True | 160 (49.5%) |
| | False [CORRECT ANSWER] | 142 (44.0%) |
| | Don't know | 21 (6.5%) |
| **Based on what you saw in the video, select True, False or Don't know/Not sure for each statement:** | | |

(*Continued*)

**Table 4.** (Continued)

|  | Statistic or Category | All Respondents (N = 323) |
| --- | --- | --- |
| ADC therapy uses the patient's T cells to fight multiple myeloma | True | 245 (75.9%) |
|  | False [CORRECT ANSWER] | 69 (21.4%) |
|  | Don't know | 9 (2.8%) |
| ADC therapy delivers medicine directly to cancer cells in your body | True [CORRECT ANSWER] | 258 (79.9%) |
|  | False | 51 (15.8%) |
|  | Don't know | 14 (4.3%) |

CAR-T = Chimeric Antigen Receptor; BiTE® = Bispecific T-Cell Engager; ADC = Anti-Drug Conjugate.

Note 1: Percentages do not include missing responses in the denominator

3. Number of survey comprehension questions answered incorrectly (*Compcount*) and the same number squared (*Compcount_squared*)

4. Time required to complete the survey in minutes (*Time*)

5. Inconsistent reporting of previous history of stem-cell transplant (*Stem*)

6. Inconsistent reporting of previous history with treatment switching (*Switch*)

**Table 5. Number of conditions reported in survey screening question.**

| Number of conditions reported | Number of respondents | Percent |
| --- | --- | --- |
| 1 | 129 | 39.94 |
| 2 | 21 | 6.50 |
| 3 | 22 | 6.81 |
| 4 | 22 | 6.81 |
| 5 | 27 | 8.36 |
| 6 | 8 | 2.48 |
| 7 | 16 | 4.95 |
| 8 | 11 | 3.41 |
| 9 | 22 | 6.81 |
| 10 | 45 | 13.93 |

**Table 6. Consistency of respondent reports.**

| Repeated question 1 | Response option | N (%) |
| --- | --- | --- |
| Have you received a stem-cell transplant to treat your multiple myeloma? | Yes | 272 (84.2%) |
| Which of the following treatments have you ever used to help manage your multiple myeloma (Select all that apply)? | Bone marrow/stem cell transplant | 211 (65.3%) |
| **Repeated question 2** | **Response option** | **N (%)** |
| Has your doctor or any other medical provider ever changed your myeloma treatment because your cancer stopped responding to treatment? | Yes | 322 (100%) |
| Have you had to change treatments for multiple myeloma? | Yes | 238 (73.9%) |

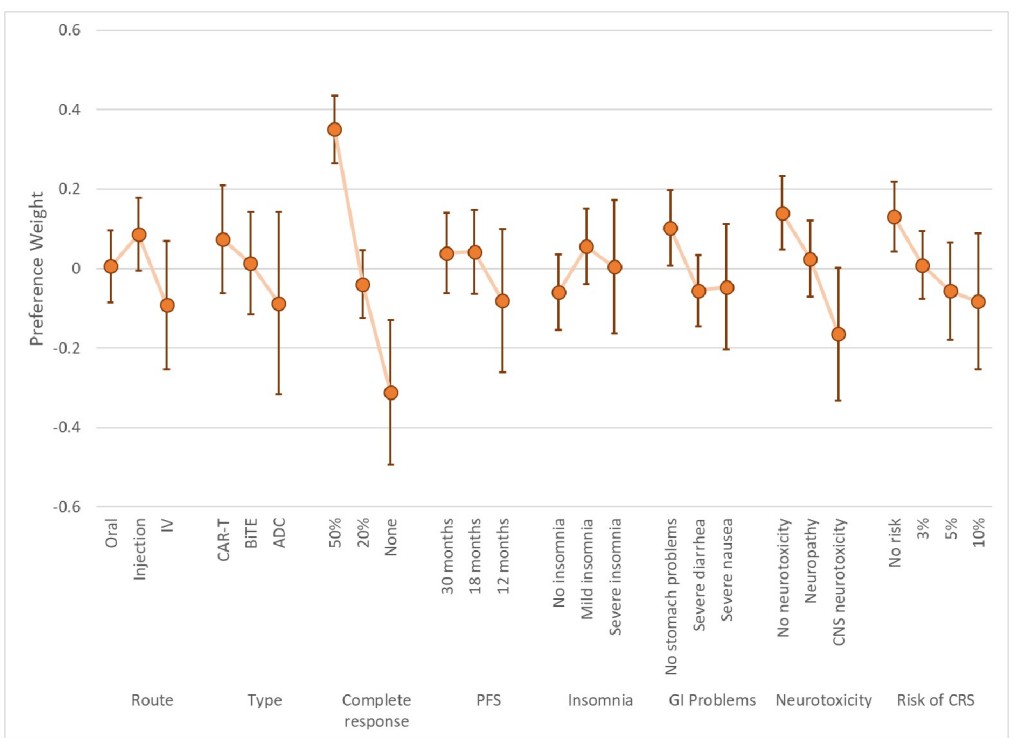

**Fig 3. Latent-class model mean preference results and 95% CIs: Class 1 (Class% = 72.1).** CIs = Confidence intervals; IV = Intravenous; CAR-T = Chimeric Antigen Receptor; BiTE® = Bispecific T-Cell Engager; ADC = Anti-Drug Conjugate PFS = Progression-free survival; GI = Gastrointestinal, CNS = Central Nervous System; CRS = Cytokine Release Syndrome. Class % is the average probability of classification into the class across respondents.

Table 7 summarizes the estimates for the impact of these characteristics on the likelihood of being in Class 1 relative to Class 2.

The marginal effects of the variables on class membership suggest that the number of diseases selected in the survey screener and the number of comprehension questions answered incorrectly by respondents were positive and significant predictors of membership for Class 2. However, the association of the number of incorrect answers with Class 2 is nonlinear and peaks around 5 out of 10 questions. After that, errors are still associated with Class 2 membership, but at a lower rate.

Fig 5 presents the overall mean preference estimates for respondents based on the weighted RPL model results. As with the LC model results, these preference weights indicate the relative importance of changes in the attribute levels. Estimates from this model are included in S1 Appendix.

## 5. Discussion

Our study looked to evaluate the quality of preference data and the potential impact of bad actors in online data collection. Importantly, we looked to identify the potential effects of bad actors on preference estimates. To do so, we used information from several background questions and general performance signals from the survey to help predict preferences.

The LC results suggest that a significant proportion of respondents provided answers with a high degree of variability and signaling perverse preferential relationships. We found that the

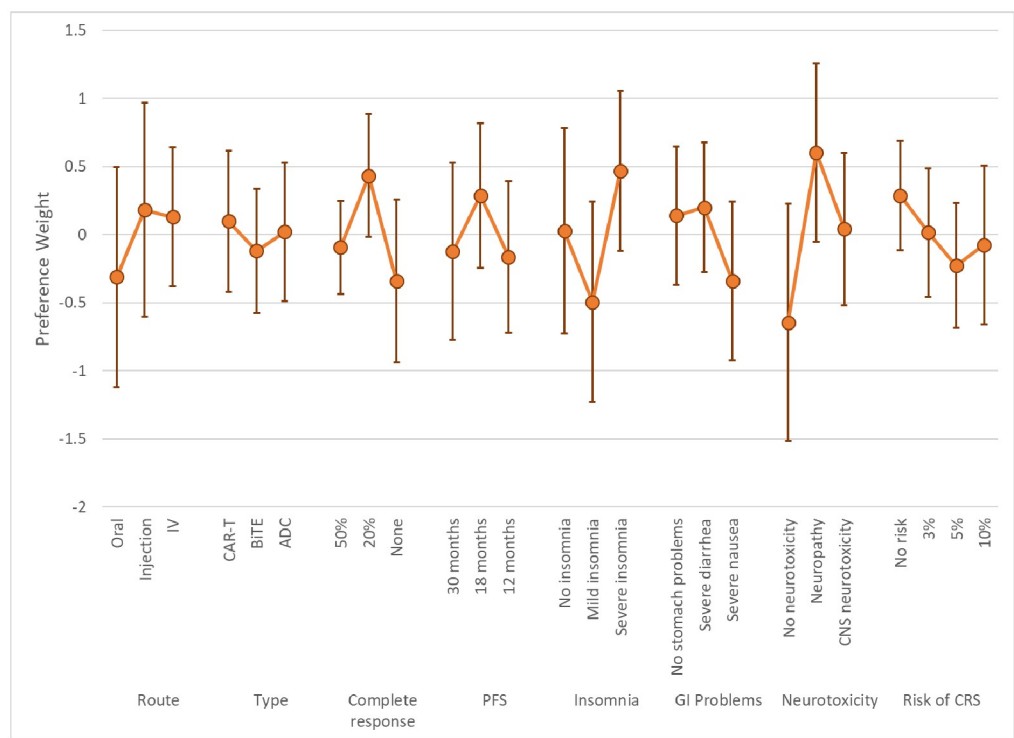

**Fig 4. Latent-class model mean preference results and 95% CIs: Class 2 (Class% = 28.9).** CIs = Confidence intervals; IV = Intravenous; CAR-T = Chimeric Antigen Receptor; BiTE® = Bispecific T-Cell Engager; ADC = Anti-Drug Conjugate PFS = Progression-free survival; GI = Gastrointestinal, CNS = Central Nervous System; CRS = Cytokine Release Syndrome. Class% is the average probability of classification into the class across respondents.

selection of multiple options in the survey screener corroborating diagnosis and frequency of incorrect answers to comprehension questions were significantly associated with membership to the respondent class with perverse and highly variable choices. While we cannot determine the extent to which respondents who exhibited these characteristics were bots or just inattentive respondents, our results support some potential hypotheses.

First, the association of multiple selection of conditions in the screener with perverse or variable choices is consistent with a pattern expected from bad actors looking to qualify for the study. That is, bots, humans or not, who could understand the role of the question as a

**Table 7. Membership estimates for being a member of Class 2 relative to Class 1.**

|  | Estimate | SE | Z-score | P>z |
|---|---|---|---|---|
| Scrcount | 0.094 | 0.046 | 2.080 | 0.038 |
| Compcount | 2.859 | 0.773 | 3.700 | <0.001 |
| Compcount_squared | -0.281 | 0.086 | 3.270 | 0.001 |
| Time | -0.054 | 0.700 | 0.080 | 0.938 |
| Age | -0.024 | 0.019 | 1.260 | 0.209 |
| Switch | -0.345 | 0.413 | 0.840 | 0.404 |
| Stem | -0.443 | 0.334 | 1.330 | 0.185 |
| Constant | 6.653 | 1.852 | 3.590 | <0.001 |

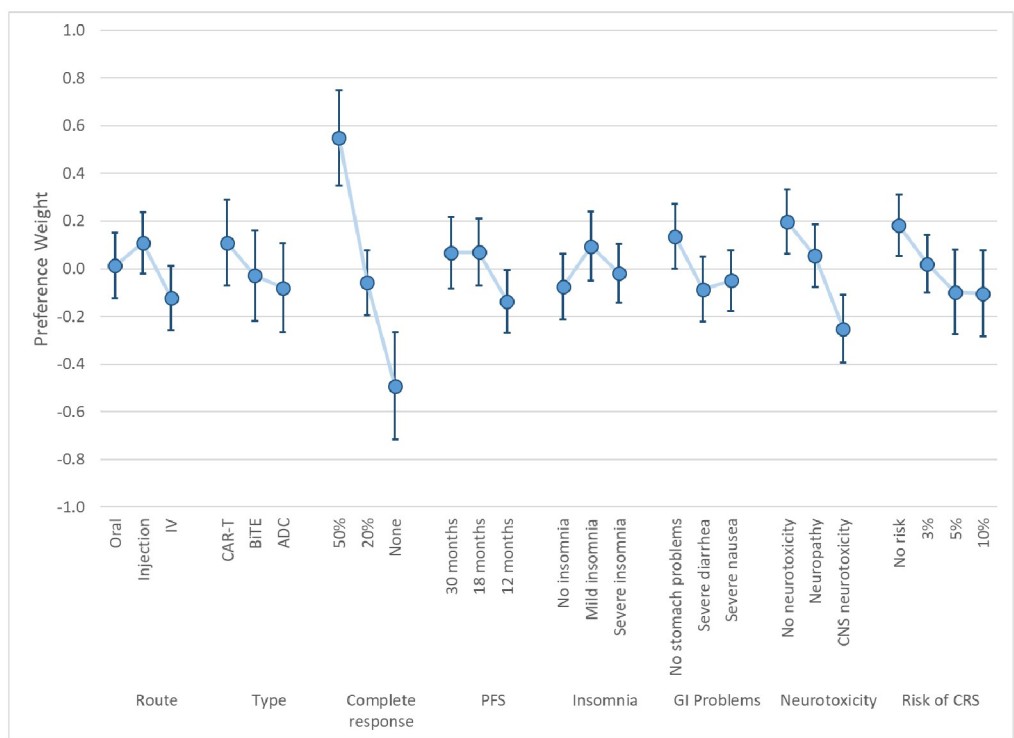

**Fig 5. Weighted RPL mean preference results with 95% CIs (N = 323).** CIs = Confidence intervals; IV = Intravenous; CAR-T = Chimeric Antigen Receptor; BiTE® = Bispecific T-Cell Engager; ADC = Anti-Drug Conjugate
PFS = Progression-free survival; GI = Gastrointestinal, CNS = Central Nervous System; CRS = Cytokine Release Syndrome.

screener, could have sought to avoid being disqualified while limiting trial and error, as this would increase the cost per completed survey.

Second, incorrect answers to comprehension questions seem to have a more nuanced effect on the identification of problematic respondents. While the general trend identified here is that wrong answers to the comprehension questions increased the likelihood of being in a problematic class, this effect peaks at 5 out of 10 questions, or 50%. Thus, suggesting that randomness in the answers to these questions is a stronger predictor of problematic measures of preferences. It is indeed possible that some respondents struggled with many comprehension questions but were able to provide systematic choices in the DCE, albeit potentially making ill-informed decisions. Those respondents would have still been included in Class 1 in our analysis.

Finally, we found no effect of completion speed on the quality of preference information. This was unexpected as the elimination of "speeders" (respondents who completed the survey too quickly) is common practice to address problematic responses [35, 36]. That said, the result is consistent with experiments relating speeding and data quality in other survey-research areas. Some of that work has found that the relationship between speeding and data quality is potentially complex and can be moderated by other respondent factors [37]. The lack of significance for this variable suggests the approach may not be as effective to limit the impact of problematic data on preference estimates.

Our work highlights the need to include multiple elements in the survey that facilitate the identification of problematic responses or respondents in the estimation of population preferences. The use of consistency checks for the DCE data may be a way to identify eccentric

responses, but without auxiliary information like the ones used here, it is likely difficult to assess whether those patterns should be excluded from or down-weighted in the population-level estimates.

More specifically, preference researchers should consider known strategies to trap bots like CAPTCHAs, impossible items, open-ended questions, and passwords. Researchers should also carefully consider the incentives provided to respondents and monitor data collection as frequently as possible [4]. We should mention that these measures could help even when working with large online consumer panels like the one we used in our study.

Our proposed strategy to evaluate the quality of responses helped us counterbalance measures that increased exposure to bad actors while reducing barriers to access a larger sample of patients. While it is not possible to determine whether the relationships uncovered here apply beyond preference research, our results offer very specific and plausible ways in which response patterns can relate to data quality. Ours is also the first study to formally define these relationships in terms of their impact on preference measures.

It is important to note that while the information collected through our survey provided an opportunity to more closely evaluate responses that did not meet a minimum level of consistency, the signals obtained from these questions must be interpreted carefully. Apparent inconsistencies could be accurate indicators of reasonable behaviors not accounted for in the study design [19]. More research is needed to understand the response patterns associated with bad actors in online preference surveys.

## 6. Conclusions

Our study highlights the need for a robust discussion around the appropriate way to handle bad actors in online preference surveys. While exclusion of survey respondents must be avoided under most circumstances, the impact of bots on preference estimates can be significant. At best, these respondents do not provide information and contribute only to the model variance. In the worst case, their responses are confounded with those collected from patients and influence the policies that are informed with preference research. The pragmatic challenges that come with online administration of surveys must be explicitly addressed to reduce the risk of compromising preference data. Not doing so can be a disservice to patients as trust in patient preference information can be undermined and more expensive data-collection approaches may be required.

## Supporting information

**S1 Appendix.**
(DOCX)

**S1 File.**
(XLSX)

## Author Contributions

**Conceptualization:** Juan Marcos Gonzalez, Thomas W. Leblanc, Bryce B. Reeve.

**Data curation:** Kiran Grover.

**Formal analysis:** Juan Marcos Gonzalez, Kiran Grover, Bryce B. Reeve.

**Funding acquisition:** Juan Marcos Gonzalez.

**Investigation:** Thomas W. Leblanc, Bryce B. Reeve.

**Methodology:** Juan Marcos Gonzalez, Bryce B. Reeve.

**Project administration:** Juan Marcos Gonzalez.

**Writing – original draft:** Juan Marcos Gonzalez, Kiran Grover, Thomas W. Leblanc, Bryce B. Reeve.

**Writing – review & editing:** Juan Marcos Gonzalez, Kiran Grover, Thomas W. Leblanc, Bryce B. Reeve.

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
