## [Decision Letter · Decision Letter 0]

27 Mar 2023

PONE-D-22-30690Did a bot eat your homework? An assessment of the potential impact of bad actors in online administration of DCE surveysPLOS ONE

Dear Dr. Gonzalez Sepulveda,

Thank you for submitting your manuscript to PLOS ONE. After careful consideration, we feel that it has merit but does not fully meet PLOS ONE’s publication criteria as it currently stands. Therefore, we invite you to submit a revised version of the manuscript that addresses the points raised during the review process. I really enjoyed reading this paper and I believe it represents a novel use of LCA to deal with the problem at hand (one that is sadly growing). I think the reviewers are both generally positive. I think the theme that underlies the reviews is that the paper could benefit from a little more clarity around the choices made in regards to the methodology. Reviewer 1 has provided a fairly comprehensive assessment of the procedures, with many good points. I think Reviewer 2 also has a useful point to make in that a less specialised reader could wonder why the DCEs themselves were about multiple myeloma, for example, and a couple of sentences explaining why this is a suitable context for your investigations would be important here. I think if you address each of the points made by the reviewers, as they are not numerous, I would be happy to give timely consideration to a revision.

We look forward to receiving your revised manuscript.

Kind regards,

Stefano Occhipinti

Academic Editor

PLOS ONE

Journal Requirements:

Financial support for this study was provided in part by Amgen, Inc. The funding agreement ensured the authors’ independence in designing the study, interpreting the data, writing, and publishing the report. 

5. We note that you have referenced (ie. Bewick et al. [5]) which has currently not yet been accepted for publication. Please remove this from your References and amend this to state in the body of your manuscript: (ie “Bewick et al. [Unpublished]”) as detailed online in our guide for authors

6. Please include your tables as part of your main manuscript and remove the individual files. Please note that supplementary tables (should remain/ be uploaded) as separate "supporting information" files

Reviewers' comments:

Reviewer's Responses to Questions

**Comments to the Author**

1. Is the manuscript technically sound, and do the data support the conclusions?

Reviewer #1: Yes

Reviewer #2: Yes

2. Has the statistical analysis been performed appropriately and rigorously? 

Reviewer #1: I Don't Know

Reviewer #2: Yes

3. Have the authors made all data underlying the findings in their manuscript fully available?

Reviewer #1: Yes

Reviewer #2: Yes

4. Is the manuscript presented in an intelligible fashion and written in standard English?

Reviewer #1: Yes

Reviewer #2: Yes

5. Review Comments to the Author

Reviewer #1: I liked this paper a lot. As a survey methodologist working in a specific field, it’s great to get evidence about a common problem from researchers in a completely different field. One problem with survey methods research is that it’s getting published across disciplines and doesn’t get read by the core survey methodologists, so kudos for sending this to PLOS-1, where it might get seen.

My comments are mainly about the presentation. I think it could be better organized to help readers like me hone in on the real issues (detailed below). I’m not offering any suggestions about the technical aspects of what they have done. I like it but these methods aren’t my specialty.

Don’t needlessly conflate mode with sample (p. 3). Online administration isn’t synonymous with opt-in sampling or even surveys with probability sampling where people in a panel or fresh recruitment can qualify to join a specialized survey by opting in, so it’s important to keep mode issues conceptually separate from sample issues. I’d suggest the authors read through the paper carefully and try to separate these issues where possible. They’ve identified a genuine problem with people insincerely trying to qualify for surveys, but this kind of thing is not an inherent problem with online surveys.

My principal suggestion is to provide a more straightforward presentation of the key issues. We learn later in the paper that bad actors opting in as MM patients are the principal problem. I think the question of how people qualify for the survey, what are the incidences, etc., should be discussed up front, since that’s where the problem gets introduced. What are the incidences in the distracter conditions in the qualifying conditions? If everyone had answered honestly, would you have screened more people to get your desired sample size? These are practical questions of interest.

One place where this focus could be implemented is on p. 8, where the authors say that “A total of 350 respondents were recruited by an international online consumer panel to complete the survey online.” But in fact far more than that were recruited, and the 350 are those who survived the screening process – which is where the biased respondents got in. I eventually figured this out, but wondering about that distracted me as I worked through the methodology of the measurement of bad behavior in the survey.

I don’t have any particular suggestions about how they identified the bad actors. This seems like an innovative approach that I’ve not seen used in other, similar studies, and I found it persuasive. Other reviewers with greater familiarity in the medical field or with latent class models may have different views.

A few small comments:

Your observation that speeding didn’t correlate with being bad actors is consistent with other research.

Are the 27 straight-liners eliminated included in the estimate of the share of the sample who were insincere or in class 2?

Reviewer #2: Dear authors, 

Thank you for the pleasure of reviewing the submitted manuscript. I have to congratulate you for producing a good manuscript. I do, however, has a few concerns. 

1) While it is clear that this research involved an experiment survey to evaluate the likelihood that bad actors can affect the quality of the data

collected, there was no justification for why you recruited patients with multiple myeloma.

I believe that this scope is more relevant to the general public, who are often contacted for public opinion surveys. The sample that you have recruited may pose a threat to the validity of the arguments that you have outlined. Therefore, I strongly suggest that you justify the reason for recruiting these patients. 

2) How do these findings extend our current understanding or current literature? For now, the manuscript informs me of the research and what has been found. It doesn't tell me how it expands on my current understanding. I cannot assess the impact of this manuscript without this information. Hence, I am unable to fully endorse it. 

These two comments are relevant to the discussion as well. I believe that if you have consolidated the introduction and the discussion, this manuscript will be an impactful research article. All the best.

6. PLOS authors have the option to publish the peer review history of their article (what does this mean?). If published, this will include your full peer review and any attached files.

Reviewer #1: No

Reviewer #2: No

---

## [Editor Report · Decision Letter 1]

13 Jun 2023

Did a bot eat your homework? An assessment of the potential impact of bad actors in online administration of preference surveys

PONE-D-22-30690R1

Dear Dr. Gonzalez Sepulveda,

We’re pleased to inform you that your manuscript has been judged scientifically suitable for publication and will be formally accepted for publication once it meets all outstanding technical requirements.

Kind regards,

Stefano Occhipinti

Academic Editor

PLOS ONE
---

## [Editor Report · Acceptance letter]

21 Jun 2023

PONE-D-22-30690R1 

Did a bot eat your homework? An assessment of the potential impact of bad actors in online administration of preference surveys 

Dear Dr. Gonzalez Sepulveda:

I'm pleased to inform you that your manuscript has been deemed suitable for publication in PLOS ONE. Congratulations! Your manuscript is now with our production department. 

Kind regards, 

on behalf of

Prof. Stefano Occhipinti 

Academic Editor

PLOS ONE